

# Loss of gut microbial diversity in the cultured, agastric fish, Mexican pike silverside (*Chirostoma estor*: Atherinopsidae)

Jesús Mateo Amillano-Cisneros[1], Perla T. Hernández-Rosas[1], Bruno Gomez-Gil[2], Pamela Navarrete-Ramírez[1,3], María Gisela Ríos-Durán[1], Carlos Cristian Martínez-Chávez[1], David Johnston-Monje[4], Carlos Antonio Martínez-Palacios[1] and Luciana Raggi[1,3]

[1] Instituto de Investigaciones Agropecuarias y Forestales (IIAF), Universidad Michoacana de San Nicolás de Hidalgo, Morelia, Michoacan, Mexico
[2] Centro de Investigación en Alimentación y Desarrollo, A.C. (CIAD), Mazatlán, Sinaloa, Mexico
[3] Cátedras-CONACYT, Consejo Nacional de Ciencia y Tecnología, Mexico City, Mexico
[4] Max Planck Tandem Group in Plant Microbial Ecology, Universidad del Valle, Cali, Valle del Cauca, Colombia

Corresponding author
Luciana Raggi,
luciana.raggi@umich.mx,
luciana.raggih@gmail.com

## ABSTRACT

Teleost fish are the most diverse group of extant vertebrates and have varied digestive anatomical structures and strategies, suggesting they also possess an array of different host-microbiota interactions. Differences in fish gut microbiota have been shown to affect host development, the process of gut colonization, and the outcomes of gene-environment or immune system-microbiota interactions. There is generally a lack of studies on the digestive mechanisms and microbiota of agastric short-intestine fish however, meaning that we do not understand how changes in gut microbial diversity might influence the health of these types of fish. To help fill these gaps in knowledge, we decided to study the Mexican pike silverside (*Chirostoma estor*) which has a simplified alimentary canal (agastric, short-intestine, 0.7 gut relative length) to observe the diversity and metabolic potential of its intestinal microbiota. We characterized gut microbial populations using high-throughput sequencing of the V3 region in bacterial 16S rRNA genes while searching for population shifts resulting associated with fish development in different environments and cultivation methods. Microbiota samples were taken from the digesta, anterior and posterior intestine (the three different intestinal components) of fish that grew wild in a lake, that were cultivated in indoor tanks, or that were raised in outdoor ponds. Gut microbial diversity was significantly higher in wild fish than in cultivated fish, suggesting a loss of diversity when fish are raised in controlled environments. The most abundant phyla observed in these experiments were Firmicutes and Proteobacteria, particularly of the genera *Mycoplasma*, *Staphylococcus*, *Spiroplasma*, and *Aeromonas*. Of the 14,161 OTUs observed in this experiment, 133 were found in all groups, and 17 of these, belonging to *Acinetobacter*, *Aeromonas*, *Pseudomonas*, and *Spiroplasma* genera, were found in all samples suggesting the existence of a core *C. estor* microbiome. Functional metagenomic prediction of bacterial ecological functions using PICRUSt2 suggested that different intestinal components select for functionally distinct microbial populations with variation in pathways related to the metabolism of amino acids, vitamins, cofactors, and energy. Our results provide, for the first time,

information on the bacterial populations present in an agastric, short-gut teleost with commercial potential and show that controlled cultivation of this fish reduces the diversity of its intestinal microbiota.

## INTRODUCTION

Gut microbiota, defined as the collective community of microorganisms that inhabit the intestine, are known to contribute to host physiology like digestion, nutrient absorption, and immune system function, significantly impacting animal health, welfare, growth, and behavior (reviewed by *Douglas, 2019*). In fish, the density, composition, and function of intestinal microbial communities are influenced by internal and external factors, including genotype, life stage, trophic level, diet, season, habitat (chemical and physical factors), sex, and phylogeny (reviewed by *Butt & Volkoff, 2019*). It is estimated that the fish gut contains between $10^7$ and $10^{11}$ bacterial cells per gram of intestinal content, and these microbes can be either residents or transients (*Nayak, 2010*; *Navarrete et al., 2012*). A total of about 145 different teleost fish species have been involved in microbiome studies, including the zebrafish model and other important fish for commercial aquaculture (*Perry et al., 2020*).

Teleosts are the most diverse group of vertebrates, occupying many different habitats, employing many varied feeding strategies, and possessing distinctive digestive specializations/anatomical configurations, which has resulted in their having a wide array of host-microbiota interactions (*Lescak & Milligan-Myhre, 2017*). For example, some agastric fish like cyprinids compensate for their lack of stomach-based acid digestion by increasing the length of their intestines and presumably depend to a greater extent on microbial activity to help break down their food (*Manjakasy et al., 2009*; *Egerton et al., 2018*). Little is known about the microbiota of agastric short-intestine teleosts lacking a true stomach, except for the zebrafish model species *Danio rerio*, which is an agastric short-intestine species of no commercial value. The study of microbial community dynamics in both wild and controlled environments studies is relatively simple with teleosts because in captivity, most fish species develop rapidly and in high quantities (*Lescak & Milligan-Myhre, 2017*). Such studies have previously demonstrated that gut microbiome composition affects host development, the process of microbiome colonization and succession, and gene-environment or immune system-microbiota interactions (reviewed by *Nayak, 2010*; *Butt & Volkoff, 2019*). As omics techniques (metagenomics, metatranscriptomics, metaproteomics, and metabolomics) continue to be developed and improved, our ability to study the ecology and function of fish microbiota is likewise evolving.

To help fill in the knowledge gap about the importance and function of agastric fish gut microbiota, we propose studying the Mexican pike silverside, *Chirostoma estor* due to its simplified digestive tract lacking a stomach and consisting instead of a short-intestine that is only 0.7 the fish's length (relative gut length - rgl) (*Ross et al., 2006*). The Mexican

pike silverside has been described by the FAO of the United Nations, as a species with high aquaculture potential due to its regional importance, low trophic level, and high nutraceutical value (high docosahexaenoic acid (DHA) content) (*Fonseca-Madrigal et al., 2014*; *Martínez-Palacios et al., 2019*; *Martínez-Palacios et al., 2020*). This teleost fish consumes mostly zooplankton and insects, which are finely ground by chewing before swallowing (*Ross et al., 2006*; *Martínez-Palacios et al., 2019*). Due to its simplified intestinal configuration, we hypothesize that the Mexican pike silverside is likely to depend more heavily on its gut microbiota for aid in digestion and nutrient absorption, compared to teleosts with stomachs or more sophisticated digestive systems. If we are to successfully develop methods of aquaculture for agastric fish such as the Mexican pike silverside, it will be important to better understand how their intestinal microbiomes are established, how these microbes influence their health and nutrition, and what conditions are necessary to optimize the interactions between bacteria and host.

In this study, we analyze and compare the gut microbiomes of *C. estor* harvested from a wild environment (Lake Patzcuaro), cultivated in indoor tanks, and raised in outdoor ponds. DNA was extracted from each fish's digesta, anterior intestine, and posterior intestine, followed by amplification of the bacterial 16S and sequencing on the Illumina Miniseq platform. To attempt predicting the metabolic activities associated with the different gut bacterial populations, we used taxonomic information to estimate metabolic functions.

## MATERIALS & METHODS

### Sample collection

Adult *Chirostoma estor* were collected from Lake Patzcuaro (LP), Mexico, in the spring of 2018 (March) (19°36′14″N, 101°37′56″W, 22 °C, altitude 2,040 masl, $N = 13$). In May of the same year, cultured adult fish of similar size to those from the lake were also harvested from fiberglass indoor tanks (tank culture = C) and outdoor earth ponds (extensive culture = E) ($N = 20$ and $N = 19$, respectively) located in the aquaculture biotechnology laboratory at Universidad Michoacana de San Nicolás de Hidalgo in Morelia, Michoacan, Mexico (19°41′22″N, 101°14′56″W, 23 °C, altitude 1,896 masl). Populations of intensively cultured fish were established twenty years ago using stock from Lake Patzcuaro. Fish from the laboratory were released into the outdoor earth ponds 8 months before sampling, and are designated as E.

At harvest, individual fish from each environment were measured and weighed (Table S1). Immediately after collection, the abdominal cavities of fish were dissected under aseptic conditions, the intestines (guts) extracted and their contents (digesta = D) were gently squeezed out and placed into sterile tubes with 96% ethanol. The guts of *C. estor* consist of a short intestine with a single loop in the midgut (Fig. S1), and were divided into two sections (anterior = A, and posterior = P gut) and placed in sterile tubes containing 96% ethanol and stored at 4 °C until later DNA extraction. All specimens were treated following EU Directives 2010/63/EU for animal experimentation and 2007/526/EC accommodation and care of animals used for experimental and other scientific purposes.

## DNA extraction and sequencing

Metagenomic DNA of fish intestines (D, A, and P) from the 3 different populations was extracted with a modified CTAB extraction protocol (*Doyle & Doyle, 1987*) by adding lysozyme (100 mg/mL), proteinase k (20 mg/mL), lithium chloride (5M), and sodium acetate (3 M, pH 5.2) to the extraction buffer. After DNA extraction, the V3 variable region of the 16S rRNA gene was PCR-amplified with the primer pair V3-338f and V3-533r (*Huse et al., 2008*). PCR products were paired-sequenced (300 cycles, 2X150) on an Illumina Miniseq, following Illumina standard protocol at Centro de Investigación en Alimentación y Desarrollo, A.C. (CIAD), Mazatlan, Sinaloa, Mexico.

## Bioinformatics

All amplicon sequence data derived from this work was submitted to NCBI Sequence Read Archive (SRA) (SRX11596261 to SRX11596334) under BioProject accession ID: PRJNA750495. After demultiplexing, CIAD sent back data as FastQ files which were preprocessed with PrinSeq (*Schmieder & Edwards, 2011*). Reads were then assembled and quality filtered with Flash v.1.2.7 software (*Magoc & Salzberg, 2011*), and VSEARCH (*Rognes et al., 2016*) was used for further processing, obtaining an abundance matrix of bacterial OTUs (operational taxonomic units clusterized at 97% identity) that was normalized using the metagenomeSeq method (*McMurdie & Holmes, 2014*). Taxonomic annotation was performed using VSEARCH (*Rognes et al., 2016*) against the SILVA138 database (*Quast et al., 2013*). The abundance matrix at the genus taxonomic level was used to calculate Good's coverage and alpha diversity indexes (*i.e.,* Chao 1 and Shannon indexes), which were estimated using the R Phyloseq library (*McMurdie & Holmes, 2013*). The genus-level abundance matrix was normalized using the metagenomeSeq method (*McMurdie & Holmes, 2014*), and the beta diversity distance matrix was calculated using Bray-Curtis dissimilarity. Beta diversity distance matrices were visualized with Non-metric Multidimensional Scaling plots (NMDS) of bacterial data grouped by environments and intestinal components, while ordination was based on between-sample dissimilarities calculated by Bray-Curtis distance and 999 permutations using library vegan 2.5-6 (*Oksanen et al., 2019*) in R software (*R Core Team, 2013*).

Although many groups define a core microbiome as those microbes with 100% occupancy across all samples (*Turnbaugh et al., 2007*), we considered (like other researchers) that a more relaxed threshold of greater than 80% occupancy would allow us to include more physiologically important bacteria in our analysis (*Baldo et al., 2015*; *do Vale-Pereira et al., 2017*; *Sweet & Bulling, 2017*; *Rimoldi et al., 2019*). Core microbiota was identified at the level of the genus when OTUs were present in at least 80% of samples per defined group (LP, C, E, D, A, and P) and at the level of OTU when present in at least one sample of each group (DLP, ALP, PLP, DC, AC, PC, DE, AE, PE). To compare microbiota between environments or digestive components, data was visualized with Venn diagrams (*Chen & Boutros, 2011*), in UpSet plots made with the package UpSetR (*Conway, Lex & Gehlenborg, 2017*).

## Statistical analysis

All statistical analyses were performed using R software (*R Core Team, 2013*) and R library vegan 2.5-6 (*Oksanen et al., 2019*). Alpha diversity and relative abundance data were tested for normality and homoscedasticity by Shapiro–Wilk's and Bartlett's tests, respectively. Data sets grouped by intestinal component (Digesta, Anterior, and Posterior gut) or environment (Lake Patzcuaro, Intensive Culture, and Extensive Culture) were found to not be normally distributed. To compare beta diversity, environmental and intestinal component groups were statistically analyzed using non-metric dimensional scaling (NMDS) based on Bray-Curtis distances (*Bray & Curtis, 1957*), Analysis of Similarities (ANOSIM), as well as Adonis tests (Permutational Multivariate Analysis of Variance, PERMANOVA) using the Bray-Curtis index at 999 permutations. The number of reads across samples was normalized by sample size, and each taxon's relative abundance (%) was calculated. Although results were generated for each taxonomic level, only Phylum and Genus levels are shown. Only taxa with the highest relative abundance were considered for statistical analysis, totaling 10 for phylum and 20 for genus. Phylum and genus abundance data were not normally distributed. Non-normal data were analyzed using nonparametric Kruskal–Wallis followed by Mann-Witney-Wilcoxon's post hoc tests with statistical significance set at $p < 0.05$. Additional analysis to detect differential abundance was performed using the Linear discriminant analysis Effect Size (LEfSe) method (*Segata et al., 2011*) integrated within the Galaxy framework (https://huttenhower.sph.harvard.edu/galaxy/). In particular, the non-parametric Kruskal-Wallis sum-rank test was used to detect differentially abundant taxa, while Linear Discriminant Analysis (LDA) was used to estimate the effect size.

## Predictive functional analysis by PICRUSt2

The functional metabolic profiles of 16S rDNA data were predicted using the Phylogenetic Investigation of Communities by Reconstruction of Unobserved States 2 (PICRUSt2) v2.3.0 beta software that uses HMMER (http://www.hmmer.org) to place OTUs into a reference phylogeny, followed by the castor R package to predict gene family abundances using hidden-state prediction. Pathway abundances are inferred based on the predicted sample functional profiles linked to pathway reactions using a modified version of MinPath (*Douglas et al., 2019*). Pathway abundance was then inferred using KEGG (Kyoto Encyclopedia of Genes and Genomes) (pathway_pipeline.py).

The output file (path_abun_unstrat_descrip.tsv) of predicted pathway abundance was loaded to R software (*R Core Team, 2013*). The 60 most abundant functional pathways (taking into account the total average of the samples) were visualized using a heatmap created by the ComplexHeatmap package in R (*Gu, Eils & Schlesner, 2016*). To determine specific differences between each of the predicted paths, the output file of PICRUSt2 (path_abun_unstrat_descrip.tsv) was loaded into the MEGAN5 package (*Huson et al., 2016*) which allowed us to define the metabolic categories predicted in these bacteria. These metabolic predictions were then analyzed using STAMP (statistical analysis of taxonomic and functional profiles) (*Parks et al., 2014*) software to further interrogate all predicted functional datasets and produce graphical depictions of key functional pathways. To

contrast group metabolic predictions by environment (LP-C-E) and intestinal component (D-A-P) samples were compared using an ANOVA and Tukey-Kramer test. Pathways with a $p$-value < 0.05 were considered to be statistically significant. For paired analysis of all nine metabolic predictions (DLP, ALP, PLP, DC, AC, PC, DE, AE, PE) simultaneously, environmental and intestinal components were compared using a two-sided Welch's $t$-test. Pathways with a $p$-value < 0.05 were considered to be statistically significant.

## RESULTS

### Bacterial diversity in the gut of *C. estor*

The bacterial community profile found in *C. estor* gut is highly variable from individual to individual, especially at the level of genus (Fig. 1A). At the level of phylum (Fig. 1B; Table S2) most samples contained Firmicutes (56.27% ± 4.28) and Proteobacteria (28.68% ± 3.47), of which the two most abundant classes were Gammaproteobacteria (24.35% ± 3.16) and Alphaproteobacteria (4.33% ± 1.25). Other common phyla were Cyanobacteria (4.69% ± 1.57), Actinobacteriota (3.51% ± 1.35), Bacteroidota (1.78% ± 0.37), Desulfobacterota (1.43% ± 0.44), Fusobacteriota (1.39% ± 1.05), Planctomycetota (0.97% ± 0.40), Verrucomicrobia (0.38% ± 0.10) and Acidobacteriota (0.29% ± 0.16).

Of the 1,207 annotated genera (Fig. 1A; Table 1), the most abundant bacterial genera observed within individual fish included *Mycoplasma* (19.95% ± 4.11), *Staphylococcus* (10.90% ± 3.30), *Spiroplasma* (10.23% ± 2.80), *Aeromonas* (7.51% ± 1.87), *Pseudomonas* (6.11% ± 1.86), *Planomicrobium* (4.75% ± 1.90), *Clostridium* (4.67% ± 1.32), *Foliisarcina* (2.37% ± 1.36) and *Cutibacterium* (2.16% ± 1.25). At the genus level, all digesta and Lake Patzcuaro samples contained significantly less *Mycoplasma* spp. than other intestinal components or environments (Table 1). Other differences were observed in the abundance of *Clostridium*, *Romboutsia*, *Prochlorococcus*, *Acinetobacter*, *Cetobacterium*, *Anaerobacter*, *Flavobacterium*, *Methylobacterium-Methylorubrum*, and *Dongia* within lake samples, while *Staphylococcus* was most abundant in the intensive culture and *Planomicrobium*, *Erwinia*, and *Dechloromonas* were most abundant in the extensive culture (Table 1).

Alpha-diversity varied considerably but not significantly by both environment and intestinal compartment, except for the Shannon index of fish digesta from Lake Patzcuaro (LP) which was significantly higher than the Shannon index of digesta from cultivated fish ($p = 0.021$) (Fig. 1C; Table S3). Bacterial diversity and Shannon index values from Lake Patzcuaro samples were significantly higher ($p = 0.00477$) than those of fish from tank cultures or outdoor ponds (Table S3).

### Environmental influence on microbial communities

NMDS of the gut microbiota in fish from different environments (Fig. 2A) revealed a nearly separate cluster of the LP samples concerning the other two environments (C and E), the latter which largely overlapped instead. These observed differences were statistically supported by ANOSIM: $R = 0.3119$, $p = 0.001$ (Fig. 2A; Table S4), and PERMANOVA (Table S4). Repeating this analysis with data from the intestinal components showed complete overlapping of the three different groups (Fig. 2B), which surprisingly appeared to

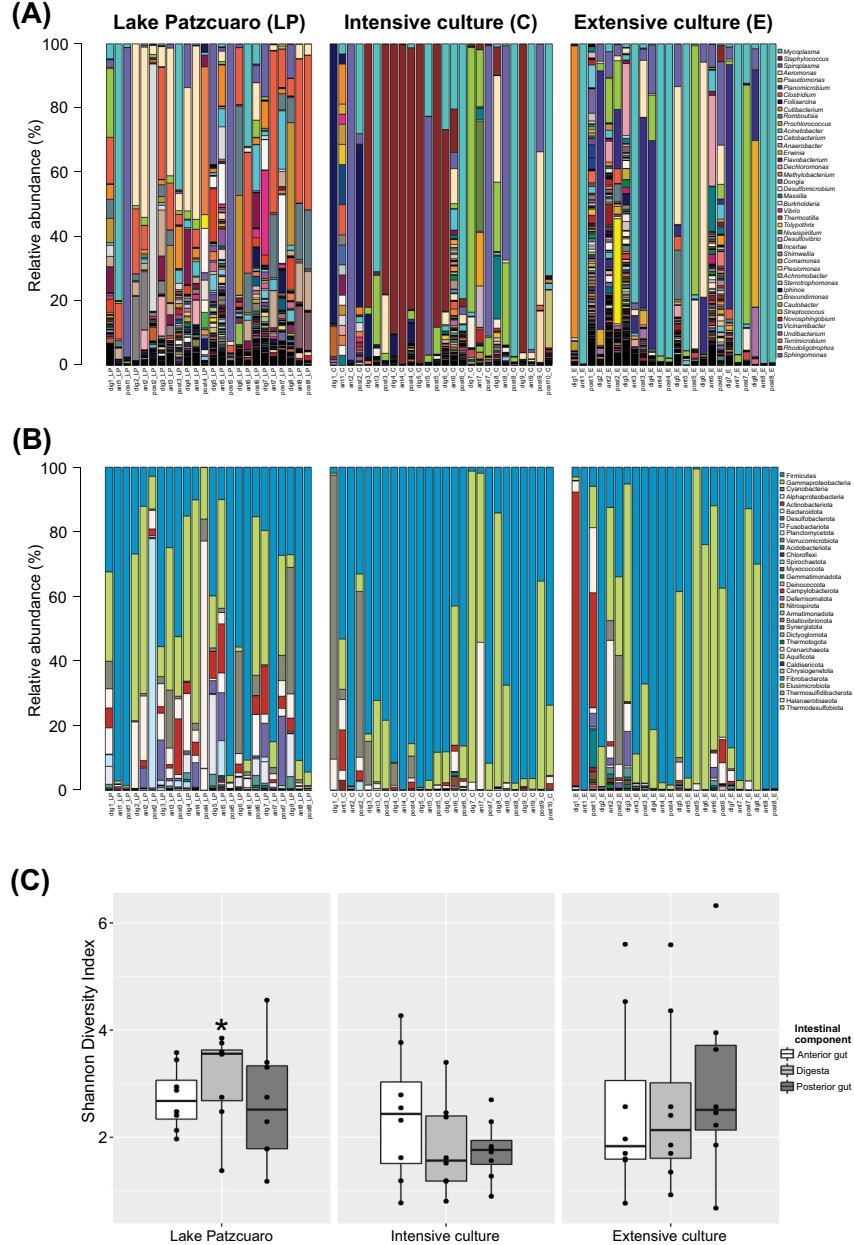

**Figure 1** **Gut microbiota diversity.** (A) Normalized profiles of bacterial genera in fish guts. (B) Normalized profiles of bacterial phyla in fish guts. Legends display only the most abundant genera phyla (with Proteobacteria classes). (C) Shannon diversity index of gut bacteria grouped by environments intestinal components, calculated using OTU data. Asterisk indicates the significant ($p < 0.05$) statistical difference between digesta from Lake Patzcuaro (DLP) and digesta from intensive culture (DC).

be significantly different from each other judging by ANOSIM (R-value of 0.06763, $p = 0.004$ (Fig. 2B; Table S5)). PERMANOVA supported this result, showing a significant difference between digesta and anterior gut with an F value of 2.2341, $p = 0.001$ (Table S5). ANOSIM and PERMANOVA analyses were stratified by intestinal component or environment,

**Table 1  Total mean relative abundance (%) (first column) ± SE, and mean relative abundance ± SE of the 20 most prevalent genera found in guts from different environments.** Data of digesta, D; anterior intestine, A; and posterior intestine, P of pike silverside in the three environments (Lake Patzcuaro, LP; Intensive Culture, C; Extensive Culture, E). Statistical comparisons were performed separately by intestinal components (D, A, P) and environments (LP, C, E). Different letters indicate statistical significance between taxonomic group abundance in a row ($p < 0.05$).

| Genus | Total abundance (N = 74) | Intestinal component | | | Environment | | |
|---|---|---|---|---|---|---|---|
| | | D (N = 24) | A (N = 25) | P (N = 25) | LP (N = 24) | C (N = 26) | E (N = 24) |
| *Mycoplasma* | 19.95 ± 4.11 | 1.55 ± 1.11 **B** | 35.88 ± 8.59 **A** | 21.68 ± 7.25 **A** | 7.17 ± 3.87 **b** | 19.94 ± 6.45 **ab** | 32.76 ± 9.31 **a** |
| *Staphylococcus* | 10.9 ± 3.30 | 18.08 ± 7.28 | 4.64 ± 3.98 | 10.28 ± 5.44 | 0.09 ± 0.03 **b** | 30.71 ± 8.14 **a** | 0.25 ± 0.21 **b** |
| *Spiroplasma* | 10.23 ± 2.80 | 5.49 ± 1.79 | 10.67 ± 5.31 | 14.35 ± 6.15 | 11.68 ± 5.47 | 14.48 ± 6.01 | 4.18 ± 1.29 |
| *Aeromonas* | 7.51 ± 1.87 | 10.62 ± 4.18 | 6.96 ± 3.23 | 5.08 ± 2.09 | 11.59 ± 3.88 | 4.72 ± 2.25 | 6.44 ± 3.48 |
| *Pseudomonas* | 6.11 ± 1.86 | 6.31 ± 3.55 | 3.93 ± 1.46 | 8.10 ± 4.14 | 1.86 ± 0.89 | 7.56 ± 3.36 | 8.79 ± 4.32 |
| *Planomicrobium* | 4.75 ± 1.90 | 11.03 ± 5.09 | 0.25 ± 0.21 | 3.23 ± 2.44 | 0.02 ± 0.01 **b** | 0.05 ± 0.04 **b** | 14.57 ± 5.39 **a** |
| *Clostridium* | 4.67 ± 1.32 | 4.18 ± 1.79 | 6.88 ± 2.94 | 2.92 ± 1.95 | 13.77 ± 3.42 **a** | 0.08 ± 0.06 **c** | 0.53 ± 0.28 **b** |
| *Foliisarcina* | 2.37 ± 1.36 | 4.55 ± 3.63 | 0.21 ± 0.13 | 2.44 ± 2.01 | 0.22 ± 0.10 **b** | 6.53 ± 3.77 **ab** | 0.02 ± 0.01 **c** |
| *Cutibacterium* | 2.16 ± 1.25 | 4.49 ± 3.77 | 1.23 ± 0.70 | 0.87 ± 0.53 | 2.21 ± 0.77 **b** | 0.63 ± 0.49 **c** | 3.78 ± 3.78 **a** |
| *Romboutsia* | 2.06 ± 0.62 | 2.68 ± 1.31 | 2.32 ± 1.11 | 1.21 ± 0.77 | 5.41 ± 1.60 **a** | 0.03 ± 0.01 **b** | 0.92 ± 0.67 **b** |
| *Prochlorococcus* | 1.43 ± 0.70 | 3.34 ± 2.00 | 0.95 ± 0.71 | 0.08 ± 0.03 | 4.25 ± 2.07 **a** | 0.01 ± 0.01 **b** | 0.14 ± 0.08 **b** |
| *Acinetobacter* | 1.37 ± 0.34 | 0.59 ± 0.14 | 1.41 ± 0.43 | 2.09 ± 0.89 | 2.66 ± 0.92 **a** | 0.68 ± 0.27 **b** | 0.83 ± 0.33 **b** |
| *Cetobacterium* | 1.37 ± 1.05 | 0.32 ± 0.16 | 0.43 ± 0.40 | 3.32 ± 3.07 | 4.05 ± 3.20 **a** | 0.11 ± 0.10 **b** | 0.04 ± 0.03 **b** |
| *Anaerobacter* | 1.17 ± 0.35 | 1.07 ± 0.62 | 1.67 ± 0.69 | 0.77 ± 0.50 | 3.49 ± 0.92 **a** | 0.01 ± 0.01 **c** | 0.11 ± 0.05 **b** |
| *Erwinia* | 0.98 ± 0.70 | 2.50 ± 2.16 | 0.18 ± 0.13 | 0.33 ± 0.20 | 0.02 ± 0.01 **c** | 0.14 ± 0.12 **bc** | 2.86 ± 2.15 **a** |
| *Flavobacterium* | 0.91 ± 0.31 | 1.50 ± 0.82 | 0.53 ± 0.17 | 0.73 ± 0.45 | 2.53 ± 0.87 **a** | 0.07 ± 0.05 **c** | 0.20 ± 0.08 **b** |
| *Dechloromonas* | 0.89 ± 0.49 | 1.14 ± 0.94 | 1.48 ± 1.15 | 0.05 ± 0.03 | 0.01 ± 0.01 **b** | 0.01 ± 0.01 **b** | 2.72 ± 1.47 **a** |
| *Methylobacterium-Methylorubrum* | 0.73 ± 0.62 | 0.02 ± 0.01 | 0.12 ± 0.05 | 2.02 ± 1.83 | 2.08 ± 1.91 **a** | 0.09 ± 0.07 **b** | 0.08 ± 0.05 **b** |
| *Dongia* | 0.71 ± 0.37 | 1.04 ± 0.84 | 0.98 ± 0.72 | 0.12 ± 0.06 | 2.17 ± 1.08 **a** | 0.01 ± 0.00 **b** | 0.01 ± 0.01 **b** |
| *Desulfomicrobium* | 0.66 ± 0.35 | 0.58 ± 0.41 | 0.53 ± 0.50 | 0.88 ± 0.85 | 1.99 ± 1.06 **a** | 0.00 ± 0.00 **b** | 0.06 ± 0.03 **a** |

showing differences in digesta taken from fish harvested from the three environments (DLP, DC, and DE), differences in anterior guts from fish harvested in ALP *vs* AC and AE, while in the posterior gut, there was a significant difference only between PLP *vs* PE (Table S5).

## Core microbiota analysis

Analysis to determine the core microbiota suggested that the genera *Mycoplasma*, *Spiroplasma*, *Aeromonas*, *Pseudomonas*, and *Acinetobacter* were present in all intestinal components of fish harvested from in all three environments (Fig. 3A). In addition to the aforementioned core genera, digesta samples from all three environments also contained *Vibrio* and *Bacillus* (Fig. 3B); all anterior intestine samples contained *Mycoplasma*, *Staphylococcus*, *Spiroplasma*, *Aeromonas*, *Pseudomonas*, *Acinetobacter*, *Vibrio*, and *Weissella* genera (Fig. 3C), while posterior intestine samples contained *Mycoplasma*, *Spiroplasma*, *Aeromonas*, *Pseudomonas* and *Acinetobacter* genera (Fig. 3D).

An UpSet plot was made to show OTUs (summed across replicates in 9 different groups) shared between the different environments and intestinal components (Fig. 3E). There was

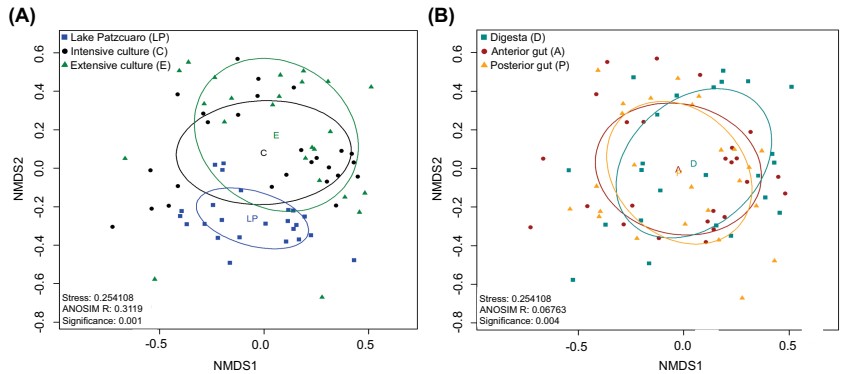

**Figure 2  Beta diversity analysis.** Nonmetric multidimensional scaling (NMDS) plot of the Bray–Curtis beta-diversity from *C. estor* microbiota profiles, estimated from the OTU abundance matrix. The two groupings were by environment (A) and by intestinal component (B).

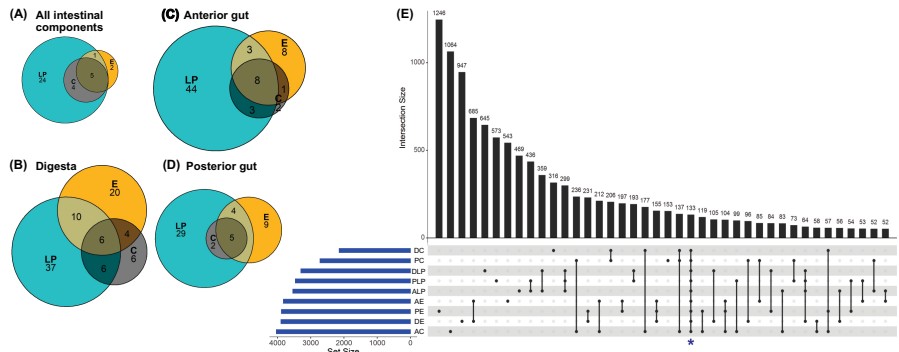

**Figure 3  Core microbiota amongst different environmental groups.** (A) Core microbiota were defined at the level of genera as present in at least 80% of the samples of a group. All three environments (LP, Lake Patzcuaro; C, Intensive Culture; E, Extensive Culture) shared five genera regardless of the intestinal compartment (5/36 genera). (B) In samples of digesta, there were six core bacterial genera (6/89 genera). (C) In samples of the fish anterior intestine, there were eight genera shared between environments (8/69 genera). (D) In samples of the fish posterior intestine, there were five genera shared between environments (5/49 genera). (E) An UpSet plot based on the presence/absence of OTUs after summing up across replicates. Presence means OTUs occurred in at least one sample of each group (DC, digesta of intensive culture; PC, posterior intestine of intensive culture; DLP, digesta of Lake Patzcuaro; PLP, posterior intestine of Lake Patzcuaro; ALP, anterior intestine of Lake Patzcuaro; AE, anterior intestine of extensive culture; PE, posterior intestine of extensive culture; DE, digesta of extensive culture; AC, anterior intestine of intensive culture). The set size of each group is plotted in horizontal bars (in blue). Bars show the number of OTUs present uniquely in a specified group and dark circles while connecting bars indicate shared OTUs between multiple samples.

a total of 133 OTUs (Fig. 3E), of which only 42 were annotated to the level of genus using the SILVA database (Table S6). The two other UpSet plots show the core microbiota consisting of 4 different genera (*Acinetobacter*, *Aeromonas*, *Pseudomona* s, and *Spiroplasma*), as well as one OTU which was present in more than 80% of all the samples (Fig. S2).

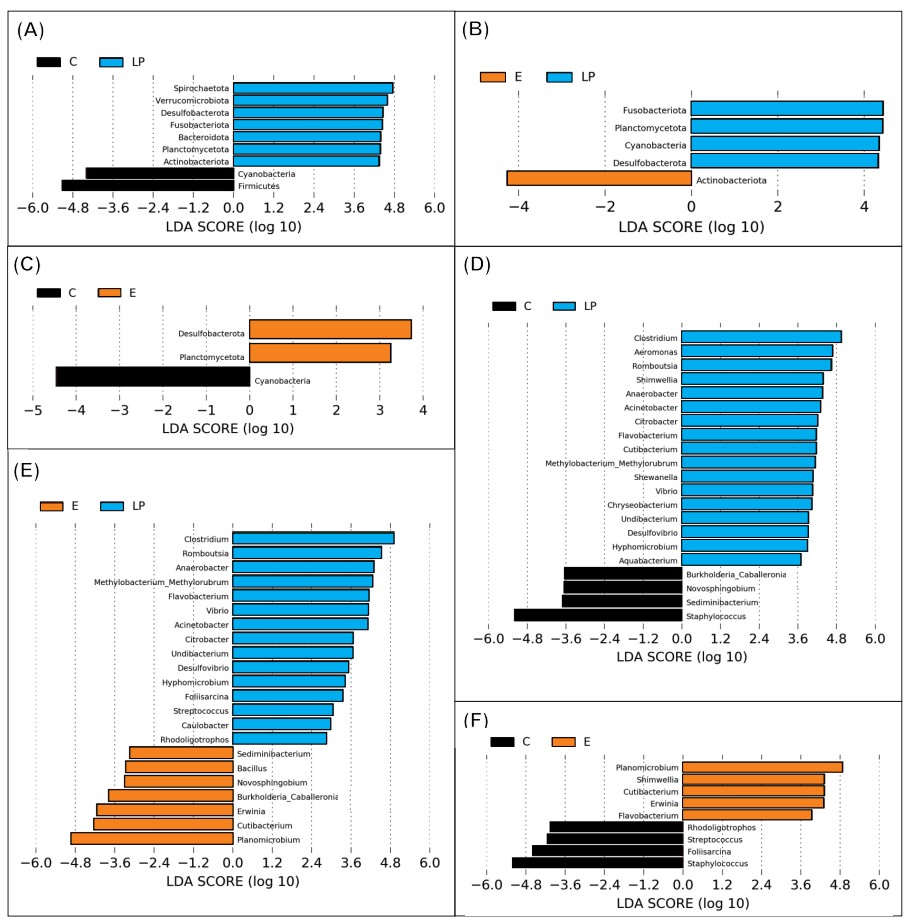

**Figure 4** **Genera prevalence deduced by LEfSe analysis.** Significant differences in the relative abundance of microbial phyla and genera by environment group (Lake Patzcuaro, LP; Intensive culture, C; Extensive culture, E). (A–C) Differences in phylum abundance between LP & C, LP & E, and C & E. (D–F) Differences in genus abundance between LP & C, LP & E, and C & E. Statistical significance of LDA effect size was evaluated in the Kruskal-Wallis test with $p$-value cutoff $= 0.05$.

## Differential abundance of taxonomic groups

A total of 11 phyla and 48 genera were present in different samples, with significant taxonomic variation when comparing LP *vs* C, LP *vs* E, and C *vs* E (Fig. 4).

LEfSe analysis showed that fish present in the LP environment had a greater diversity of phyla than fish grown in C (7 *vs* 2) or E (4 *vs* 1) and that in general, fish grown in the E environment had a greater diversity of phyla than fish grown in the C environment (2 *vs* 1) (Figs. 4A, 4B and 4C). Zooming into the level of genus, there were 17 and 15 genera that were more highly abundant in LP than in C (4) or E (7) groups, respectively (Figs. 4D and 4E). The comparison between C or E fish microbiota suggests that only 4 or 5 genera were abundant in each (Fig. 4F).
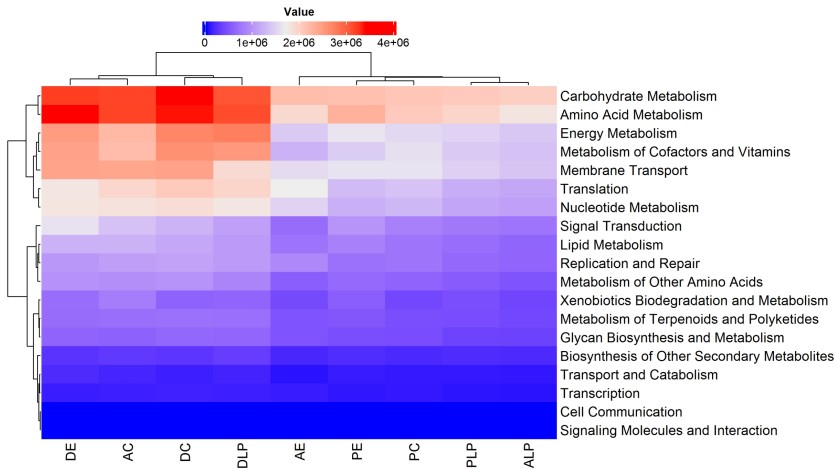

**Figure 5  Heatmap and dendrogram of KOs pathways in nine different groups.** Prediction by PICRUSt2 is shown at KEGG's first level. AE, anterior intestine/extensive culture; ALP, anterior intestine/Lake Patzcuaro; PLP, posterior intestine/Lake Patzcuaro; PC, posterior intestine/intensive culture; PE, posterior intestine/extensive culture; DLP, digesta/Lake Patzcuaro; DC, digesta/intensive culture; AC, anterior intestine/intensive culture; DE, digesta/extensive culture.

## Functional metabolic prediction using PICRUSt2

A total of 440 functional pathways were predicted for the 74 different samples (from all components and environments) that were analyzed. Of these, the 60 most abundant and with statistical differences between samples were re-analyzed, with metabolic predictions shown in Fig. S3. It was predicted that certain metabolic pathways would be over-represented in digesta samples relative to others (DLP, D, and DE). The majority of the functional KO pathways predicted to occur in all 9 groups belonged to four main categories: (i) Metabolism, (ii) Genetic Information Processing, (iii) Environmental Information Processing, and (iv) Cellular Processes (Fig. 5).

Within these predicted metabolic pathways, genes associated with the metabolism of energy, cofactors, vitamins, nucleotides, carbohydrates, and amino acids were overrepresented in cultured anterior intestines and digesta samples harvested from all environments (DLP, DC, AC, and DE) (Fig. 5). Predicted differences in metabolic pathways and their regulation are shown in Figs. S3 and S4.

## DISCUSSION

Gut microbiota has the potential to influence the health and welfare of fish raised in commercial aquaculture, making it important to understand what factors may influence microbiome diversity. Hypothesizing that captive conditions would alter the gut microbiome of the Mexican pike silverside, we compared the microbiota in wild fish to those raised in captivity. We characterized the microbiota found in the digesta and intestines of *C. estor* collected from different environments using high-throughput DNA sequencing.

The most abundant bacterial phyla in *C. estor* guts harvested from all three environments were Proteobacteria and Firmicutes, which are both common and abundant in most fish species (*Egerton et al., 2018*). Despite numerous published studies on fish microbiomes, few publications report their results at the genus level as we have here in the current study. In agreement with other freshwater teleosts, the following genera were abundant in many of our samples: *Propionibacterium* (or *Cutibacterium*), *Staphylococcus*, *Clostridium*, *Flavobacterium*, *Cetobacterium*, *Pseudomonas*, *Romboutsia*, *Spiroplasma*, *Vibrio*, *Aeromonas*, and *Mycoplasma* (*e.g.*, *Roeselers et al., 2011*; *Gajardo et al., 2016*; *do Vale-Pereira et al., 2017*; *Rimoldi et al., 2019*; *Ruzauskas et al., 2021*). Although it is hard to predict physiological roles for these microbes, some bacteria of the genera *Staphylococcus*, *Vibrio*, and *Aeromonas* can be opportunistic pathogens, while others have been used as probiotics in fish diets promoting positive immune and growth responses (*Austin et al., 1995*; *Taoka et al., 2006*; *Pieters et al., 2008*; *Abd El-Rhman, Khattab & Shalaby, 2009*; *Korkea-aho et al., 2012*; *Gao et al., 2016*; *Qi et al., 2020*; *Tran et al., 2020*). *Undibacterium* spp. was abundant in wild *C. estor*, and has also been isolated from zebrafish (*Danio rerio)* and Korean shiner (*Coreoleuciscus splendidus)* (*Kämpfer et al., 2016*; *Lee et al., 2019*). Wild *C. estor* contained abundant *Anaerobacter*, *Methylobacterium-Methylorubrum*, and *Hyphomicrobium*, and while their influence on fish physiology is unknown, they are also known to be abundant in other aquatic organisms such as coral (*Maire et al., 2021*).

Interestingly, our results suggest that the microbiota of *C. estor* is very similar to that of other freshwater omnivorous fish (*Sullam et al., 2012*), containing both high levels of Actinobacteria while lacking Bacteroidetes. *C estor* also contained a high relative abundance of Firmicutes and Cyanobacteria, a low relative abundance of Proteobacteria, and an absence of Enterobacteriales which are strongly characteristic of herbivorous marine fish (*Sullam et al., 2012*). These particularities of the *C. estor* microbiome might be diagnostic for a fish with a specialized zooplanktonic and planktonic diet and may be linked to its specialized form of agastric digestion.

*C. estor* gut samples had high inter-individual variability (Figs. 1A and 1B), which could reflect the stochastic instability of microbial diversity in the rapidly changing environment of the short *C. estor* intestine. Despite this high variability, we were able to predict the existence of a core microbiome in *C. estor* (Figs. 3A–3D and Fig. S2 above). Core microbiota included 4 genera and 133 OTUs, which is a number similar to the core microbiota for other fish species (*Dehler, Secombes & Martin, 2017*; *Rimoldi et al., 2019*).

Most significantly, the diverse bacterial genera observed exclusively in Lake grown fish samples (24 genera and 166 OTUs), suggests that fish grown in captivity lose some of their microbial diversity. Future experiments will be required to study the impact that this loss of bacterial diversity might have on fish digestion, health, and growth. Similar to other published experiments and aiming to find bacteria with probiotic potential, such experiments might include transplanting of microbial consortia or cross inoculation with individual strains isolated from the intestines of wild fish, farmed fish, or autochthonous aquatic microbial communities (*Robertson et al., 2000*; *Wang & Xu, 2006*; *Abd El-Rhman, Khattab & Shalaby, 2009*; *Essa et al., 2010*; *Korkea-aho et al., 2012*; *Dias et al., 2018*; *Li et al., 2019*; *Mohammadian et al., 2019*; *Mukherjee, Chandra & Ghosh, 2019*; *Xia et al., 2020*).

Identification of a core microbiota and the results of the differential abundance analysis (LEfSe) showed that samples from Lake Patzcuaro had higher prevalent and abundant genera compared to fish samples from Intensive and Extensive Cultures. This could be explained by the zooplanktonic feeding habits of the species (*Martínez-Palacios et al., 2007*) and the greater variety of food items found in the wild. Our results are in line with the previously published theory that a wild environment is a greater source of bacterial diversity than a controlled environment (*Dehler, Secombes & Martin, 2017*).

Bacterial diversity seemed to be higher in the anterior *vs* the posterior intestine (8 *vs.* 5 core genera), which could be related to differences in digestive physiology (pH) along the digestive canal as has been reported previously (*Martínez-Palacios et al., 2002*). Nevertheless, individual gut components didn't develop significantly distinct microbial communities (Fig. 2B), explained perhaps by the constant turnover of intestinal contents in fish that feed very frequently.

Prediction of differential microbiome metabolic potential using PICRUSt2 suggested that genes associated with amino acid, vitamins and cofactors, and energy metabolic pathways would be over-represented in certain compartments of the digestive system in *C. estor*, implying (as also observed by others for other fish species) a link between microbial diversity and fish physiological function (*Geraylou et al., 2012*; *Li et al., 2017*; *Agus, Planchais & Sokol, 2018*). It is important to remember that results from PICRUSt2 are only bioinformatic predictions based on the taxonomy of 16S rRNA genes and should be interpreted with caution. It will be important to follow up our experiment with more in-depth functional studies such as metatranscriptomics, metabolomics, and microbiome transplantation to validate predictions about microbiota functionality in this fish species.

## CONCLUSIONS

This is the first study on the Mexican pike silverside *Chirostoma estor* (a short-gut agastric model) to employ high-throughput sequencing of 16S rRNA genes to define the diversity of intestinal bacteria while attempting to predict their function. Microbial diversity was lowest in fish cultured in fiberglass tanks, intermediate in fish cultured in earth ponds, and highest in wild fish. These results suggest that husbandry can influence Mexican pike silverside microbiota, with possible implications for its commercial aquaculture. Restoration or optimization of lost intestinal microbiota, perhaps through autochthonous probiotic diet supplementation, may become a technology to promote better growth and health of captive fish in the future.

Microbial profiles showed high inter-individual variation that could be due to the dynamic nature of this fish's digestive tract. Finally, a core microbiota was identified in *C. estor*, although few of the genera identified were similar to those of other fish species. Compared to other fish species, this agastric short-intestine fish also possesses a unique digestive microbiota. The development of culturing methods of similar agastric fish with aquaculture potential such as anchovies, sardines, other atherinopsids, and hemyranphids may benefit from this intestinal microbiome data on the model agastric fish *Chirostoma estor*.

## ACKNOWLEDGEMENTS

The authors thank MSc Sibila Concha Santos and Jesús López García for their invaluable technical assistance in the LANMDA laboratory and the sampling field.

### Funding

This study was supported by Laboratorio Nacional de Nutrigenómica y Microbiómica Digestiva Animal and Consejo Nacional de Ciencia y Tecnología (projects No. 315841 and 315209), Coordinación de la Investigación Científica-UMSNH, and CONACYT doctoral scholarship 701910. The funders had no role in study design, data collection and analysis, decision to publish, or preparation of the manuscript.

### Grant Disclosures

The following grant information was disclosed by the authors:
Laboratorio Nacional de Nutrigenómica y Microbiómica Digestiva Animal and Consejo Nacional de Ciencia y Tecnología (projects No. 315841 and 315209).
Coordinación de la Investigación Científica-UMSNH.
CONACYT doctoral scholarship 701910.

### Competing Interests

The authors declare there are no competing interests.

### Author Contributions

- Jesús Mateo Amillano-Cisneros performed the experiments, analyzed the data, prepared figures and/or tables, authored or reviewed drafts of the paper, and approved the final draft.
- Perla T. Hernández-Rosas performed the experiments, analyzed the data, prepared figures and/or tables, and approved the final draft.
- Bruno Gomez-Gil and Pamela Navarrete-Ramírez conceived and designed the experiments, authored or reviewed drafts of the paper, and approved the final draft.
- María Gisela Ríos-Durán and David Johnston-Monje analyzed the data, authored or reviewed drafts of the paper, and approved the final draft.
- Carlos Cristian Martínez-Chávez conceived and designed the experiments, performed the experiments, authored or reviewed drafts of the paper, and approved the final draft.
- Carlos Antonio Martínez-Palacios conceived and designed the experiments, performed the experiments, analyzed the data, authored or reviewed drafts of the paper, and approved the final draft.
- Luciana Raggi conceived and designed the experiments, performed the experiments, analyzed the data, prepared figures and/or tables, authored or reviewed drafts of the paper, and approved the final draft.

## Animal Ethics

The following information was supplied relating to ethical approvals (i.e., approving body and any reference numbers):

Consejo Nacional de Ciencia y Tecnologia (CONACYT), Mexico, provided full approval for this research (2018-294826).

## DNA Deposition

The following information was supplied regarding the deposition of DNA sequences:

The sequence data are available at the NCBI BioProject: PRJNA750495.

## Data Availability

Raw sequence data are available at the NCBI Sequence Read Archive (SRA) (SRX11596261 to SRX11596334).

## Supplemental Information

Supplemental information for this article can be found online at http://dx.doi.org/10.7717/peerj.13052#supplemental-information.

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
