# Peer review of "Loss of gut microbial diversity in the cultured, agastric fish, Mexican pike silverside (Chirostoma estor: Atherinopsidae)"

_PeerJ, doi:10.7717/peerj.13052_

## Round 0.1 · original submission · Minor Revisions

As stated by the reviewers, the current manuscript shows promising results, especially in the changes of gut microbiota in teleosts after domestication. However, I hope the authors would be able to address some concerns raised by both reviewers before the manuscript is acceptable.

Reviewer 1 ·

Basic reporting

The manuscript reports on the gut microbiota of a freshwater carnivorous species from Mexico. This species has potential for aquaculture

Abstract: it will be nice if the authors can insert the actual problem statement/hypothesis of their research. Although it is mentioned at the end of abstract that this is potentially the first microbiota of a short-gut teleost, it will be proper to mention in abstract the status of gut microbiota studies in teleost, with regards to the different gut morphology or structure.

Introduction

-Paragraphs 1-4 should be edited into two paragraphs. One should deal with the concept of gut microbiota. The other should provide a good background on fish gut microbiota research- the trend, design, progress, leading to the reasons for this study.

I personally feel there are a lot of references cited in these 4 paragraphs but not enough information (important ones). The authors should try and select the important ones and state them as part of the Introduction text

Experimental design

Ln 104-105: The fish samples had an average weight and a total length of 27 g (11-52 g) and 15 cm (11-18 cm), respective – which cohort are these? Wild from lake or biotechnology lab or ponds? Its not clear

Ln 119- please add `from the 3 different populations’

Validity of the findings

Discussion

Ln 298-299: it is not true that most studies on fish microbiota only report on phyla levels. Please edit this sentence. Even the authors compared their findings are genus level with a whole series of other studies in line 303 onwards, a list of citations were provided.

Ln 300: ‘Such studies report that fish present most abundantly Proteobacteria and Firmicutes phyla at different proportions’ – please edit this sentence


Conclusion
What about the novelty in terms of microbiota of short gut fish with other species? for example, comparison with herbivorous species. This is also not discussed in Discussion

Reviewer 2 ·

Basic reporting

The manuscript is structured appropriately and contains appropriate figures. The authors include relevant references and context.

The sequencing data is available through NCBI with metadata appended. It may helpful to add more specific metadata explicitly tying samples to the environment types (rather than relying on lat long) but that is not absolutely necessary.

The English language should be improved to ensure that an international audience can clearly understand the text. Places where the language could be improved include (but are not limited to) lines 54, 72-73, 109-110, 178-180, 313-317, 321-324 as your current phrasing makes comprehension difficult. I would suggest you have a colleague who is proficient in English and familiar with the subject matter review your manuscript, or contact a professional editing service.

Experimental design

The authors profiled gut microbial communities from C. estor looking for variation between three environments and three gut compartments. It is an interesting addition to the literature on fish microbiota. I do have a couple of concerns about the presentation (discussed here) and analysis (presented in the next section) that could be addressed to improve the manuscript.

Presentation concerns
Environments:
The environment types are sometimes described as representing different domestication states (e.g. line 30) but the authors do not provide any information on the evolutionary history of these populations to support that claim. I believe it is interesting to analyze these three populations but they must be discussed appropriately. Unless you can demonstrate that the captive populations have been under selection for tameness and tolerance of captive environment, there is not cause to call them domesticated. I would instead present as wild or free-living and cultivated or captive.
Among the captive populations, the difference between intensive and extensive culture (e.g. line34) is unclear. I think this is mostly because “extensive” is not an intuitive word to use here. Maybe use “cultivated indoor” and “cultivated outdoor”?
Regardless of what you call them, it would be helpful to provide more information on the history of these populations in lines 99-104. For instance, how long have the cultivated populations been in cultivation for?

Gut sections:
You refer to the different gut sections as both components and compartments. Be consistent throughout. It would be great if you could provide a figure of the gut section breakdown to help reader understand order and relative size.

Language edits:
-It would be helpful if you were consistent in what you called the environments and whether you use abbreviations (e.g. line 243) or something else.
-line 128, 130, etc. these are more accurately described as amplicons rather than metagenome sequences since they are the result of PCR of a target region.
-line 178-180 It is unclear what you mean here. To be determined “common” must an OTU present in all samples or in 80%?
-line 235 what are overall samples?
-line 236, 238 etc. can you provide the exact p value instead of just <0.05?
line 237-239 does this mean wild is higher than both cultivated populations together or both analyzed separately?

Figures:
The Venn diagram should be scaled so sectors with more taxa included are larger, otherwise it is just a colorful table.

Validity of the findings

Analysis concerns

Differential abundance analyses:
You provide lefSe and edgeR analyses to do differential abundance tests but don't articulate what distinguishes them. If you can't say the strengths provided by both together, I would just include one. Since lefSe only compares 2 groups at a time and your variables have 3 levels, it seems best to not include it.
It may be interesting to add an analysis of what is differentially abundant between wild and cultivated (grouping intensive and extensive together) to see what the effect of the general environment is. This would allow you to more robustly say what is different about the wild population.
For any differential abundance analysis you should specify what sort of multiple hypothesis correction was carried out.
It appears differential abundance results are reported both in lines 227-233 and 265-277. Or was some other set of tests conducted for 227-233, if so why include both?

Other statistics:
For the analysis of gut sections, you should be controlling for individual since they are not independent if from the same gut. You can do this by stratifying in anosim or with a random effect in some differential abundance models.
You describe both ANOSIM and PERMANOVA In the methods (lines 152-154) but only ANOSIM appears in the results. They are mostly interchangeable so I see no need to include both.
Overall, it would be helpful if the analysis code, including the processing of sequencing reads, was made available in supplement or on GitHub or an equivalent server.

---

## Round 0.2 · Minor Revisions

The revised version of the manuscript sufficiently addressed the comments from the reviewers, and the results and discussion are solid. However, I would like to request the authors to either seek proofreading service or professional aid in revising the grammar/language of the whole manuscript for consistency.

---

## Round 0.3 · Minor Revisions

I regret to inform you that the English language of the manuscript is still in need of improvement and editing.

---

## Round 0.4 · accepted · Accept

Upon reviewing the latest version of the manuscript, I recommend its acceptance in its current form. Congratulations to the authors!